# Factors Influencing the Repeated Transient Optical Droplet Vaporization Threshold and Lifetimes of Phase Change, Perfluorocarbon Nanodroplets

**DOI:** 10.3390/nano13152238

**Published:** 2023-08-02

**Authors:** Andrew X. Zhao, Yiying I. Zhu, Euisuk Chung, Jeehyun Lee, Samuel Morais, Heechul Yoon, Stanislav Emelianov

**Affiliations:** 1Wallace H. Coulter Department of Biomedical Engineering, Georgia Institute of Technology, Emory University School of Medicine, Atlanta, GA 30332, USA; azhao36@gatech.edu; 2School of Electrical and Computer Engineering, Georgia Institute of Technology, Atlanta, GA 30332, USAeuisuk.chung@gatech.edu (E.C.); jeehyunlee@gatech.edu (J.L.); smorais@gatech.edu (S.M.); 3School of Electronics and Electrical Engineering, Dankook University, Yongin-si 16890, Republic of Korea; heechul.yoon@dankook.ac.kr

**Keywords:** perfluorocarbon nanodroplets, emulsion, photoacoustic, ultrasound, vaporization, ODV

## Abstract

Perfluorocarbon nanodroplets (PFCnDs) are sub-micrometer emulsions composed of a surfactant-encased perfluorocarbon (PFC) liquid and can be formulated to transiently vaporize through optical stimulation. However, the factors governing repeated optical droplet vaporization (ODV) have not been investigated. In this study, we employ high-frame-rate ultrasound (US) to characterize the ODV thresholds of various formulations and imaging parameters and identify those that exhibit low vaporization thresholds and repeatable vaporization. We observe a phenomenon termed “preconditioning”, where initial laser pulses generate reduced US contrast that appears linked with an increase in nanodroplet size. Variation in laser pulse repetition frequency is found not to change the vaporization threshold, suggesting that “preconditioning” is not related to residual heat. Surfactants (bovine serum albumin, lipids, and zonyl) impact the vaporization threshold and imaging lifetime, with lipid shells demonstrating the best performance with relatively low thresholds (21.6 ± 3.7 mJ/cm^2^) and long lifetimes (t_1/2_ = 104 ± 21.5 pulses at 75 mJ/cm^2^). Physiological stiffness does not affect the ODV threshold and may enhance nanodroplet stability. Furthermore, PFC critical temperatures are found to correlate with vaporization thresholds. These observations enhance our understanding of ODV behavior and pave the way for improved nanodroplet performance in biomedical applications.

## 1. Introduction

Perfluorocarbon nanodroplets (PFCnDs) were developed as an alternative ultrasound contrast agent to microbubbles. These particles are synthesized at small sizes to allow them to extravasate and reach extravascular targets that are beyond the reach of conventional microbubbles [1,2,3,4,5,6]. Upon reaching the targeted location, these nanodroplets can then be stimulated acoustically or optically to form a microbubble in situ. This ability to extravasate and be remotely stimulated allows for the nanodroplets to be used for extravascular applications such as examining lymph node trafficking and cancer treatment through drug delivery and tissue occlusion [2,7,8]. Furthermore, the nanodroplet vaporization itself can be utilized for other applications such as blood-brain barrier opening and cell sonoporation [9,10,11,12].

To optically stimulate these nanodroplets, a photoabsorber, such as a dye or nanoparticle, is incorporated into the nanodroplet [13]. Optical stimulation allows for simultaneous monitoring of the vaporization without any acoustic interference. This permits the detection of photoacoustic contrast and can be combined with high-frame-rate ultrasound imaging to visualize the vaporization and recondensation of PFCnDs formulated with perfluorocarbons that have boiling points higher than physiological temperatures [14]. This combination can allow for super-resolution [15], background-free imaging [16], multiplexing through the use of multiple dyes [8], enhancement of vaporization dynamics through ultrasound [17], and image-guided drug delivery [18]. However, the ODV fluence threshold of sub-micrometer droplets has not been well characterized due to the use of a variety of different absorbers and synthesis methods. It remains unclear what factors influence the laser fluence required for vaporization.

In contrast, acoustic droplet vaporization (ADV) has been extensively characterized. Various factors related to the droplets, such as droplet size, surface tension, droplet concentration, and core boiling temperature, along with ultrasound activation parameters, including frequency, pulse repetition frequency, number of cycles, and pressure, have been found to impact the ADV threshold [19,20,21,22]. Some studies have shown similar dependencies in ODV thresholds. For instance, ODV thresholds depend on the boiling temperature of the PFC core, where droplets containing octafluoropropane (boiling point: −36.7 °C) possess a ninefold lower vaporization threshold than PFP (boiling point: 28 °C) [4]. However, it is unclear how similar ODV is to ADV. 

Most relevant ODV work has examined larger droplets (~1 µm diameter) to characterize the underlying dynamics and physics of vaporization and utilized optical methods to observe the vaporization [6,23,24,25]. Although studying larger droplets can provide insights into the vaporization phenomenon, these droplets are not within the clinically relevant size range capable of extravasation, and the observed behaviors may not translate to smaller sizes. Thus, it is important to characterize the vaporization thresholds of smaller droplets (~300 nm in diameter) to properly engineer their properties.

Furthermore, previous works have primarily characterized the vaporization thresholds optically. This approach is unable to visualize transient vaporization behavior at the sub-micron scale, as the phenomenon occurs on the order of micro- to milliseconds and may not have any relevance for imaging within the body [17,23]. In this paper, we take advantage of high frame rate ultrasound to visualize optically triggered vaporization and characterize which factors, such as shell material, environment, laser pulse repetition frequency (PRF), and nanodroplet core, influence the vaporization threshold of clinically relevant-sized nanodroplets.

## 2. Materials and Methods

### 2.1. Nanodroplet Synthesis and Characterization

Nanodroplets with different shells were synthesized using probe sonication with the same settings among all the different formulations (Figure 1).

Lipid droplets were synthesized following methods previously described [26]. Briefly, DSPE-PEG 2000 (1,2-distearoyl-sn-glycero-3-phosphoethanolamine-N-[methoxy(polyethylene glycol)-2000]; Avanti Polar Lipids, Inc., Alabaster, AL, USA) and 18:0 PC (DSPC) (1,2-distearoyl-sn-glycero-3-phosphocholine; Avanti Polar Lipids, Inc., Alabaster, AL, USA) (9 DSPE-PEG 2000:1 DSPC by mass) in chloroform were mixed with 1 mg of dye Epolight 3072 (Epolin). The chloroform was then evaporated using a rotary evaporator to form a lipid cake. Phosphate buffered saline (PBS, 2 mL) was added to the flask and bath sonicated to resuspend the lipid and dye. The solution was then transferred to an 8 mL glass vial, and perfluorocarbon liquid (100 µL of perfluoropentane (PFP) and perfluorohexane (PFH) from FluoroMed (Round Rock, TX, USA) and perfluoroheptane (PFHept) from Sigma-Adlrich, St. Louis, MO, USA) was then added, and then probe sonicated (Q700, QSONICA, Newtown, CT, USA) in an ice bath at an amplitude of 1 (30 watts/cm^2^) for a processing time (total on time as defined by the manufacturer) of 20 s with a pulse-ON time of 1 s and a pulse-OFF time of 5 s followed by a second sonication sequence at an amplitude of 50 (94 watts/cm^2^) for a processing time of 5 s with a pulse-ON time of 1 s and a pulse-OFF time of 10 s. 

The nanodroplet solution was then centrifuged at 300 RCF for 2 min to separate out the sub-micron droplets from the larger droplets and unincorporated dye. The supernatant was then centrifuged at 3000 RCF to pellet all the nanodroplets and separate them from unincorporated lipids. The nanodroplets were then resuspended in 2 mL of PBS.

Zonyl droplets were synthesized by adapting methods described elsewhere [9]. Essentially, epolight 3072 (1 mg, Epolin, Newark, NJ, USA) was mixed with 2 mL of 0.25% zonyl solution (Sigma-Aldrich, St. Louis, MO, USA). The solution was bath sonicated to ensure uniform distribution of the dye. Then, PFH (100 µL) was added, and the solution was probe-sonicated in an ice bath using previously described settings. After sonication, the solution was centrifuged at 300 RCF for two minutes to remove unincorporated dye and large droplets, and then centrifuged at 3000 RCF to remove unincorporated surfactant and dye. Bovine serum albumin (BSA, Sigma-Adlrich, St. Louis, MO, USA) droplets were synthesized using similar methods, except the zonyl solution was replaced with a BSA solution (2 mL, 2 mg/mL) and 1 mg of Epolight 3072 [27]. The remaining protocol was the same as described above. Dynamic light scattering (DLS, Zetasizer Nano ZS, Malvern Panalytical, Malvern, UK) and nanoparticle tracking analysis (NTA, Nanosight NS300, Malvern Panalytical, Malvern, UK) were used to determine the size and concentration measurements. Nanodroplets were diluted based on concentration measurements to concentrations of 5 × 10^9^ nanodroplets/mL. 

In order to characterize the nanodroplet size and concentration change over the laser pulse, the lipid, PFH, nanodroplets were diluted 200-fold and placed in a 96-well plate. Using a custom-built well plate lasing system, the nanodroplets were lased at 75 mJ/cm^2^ for various numbers of pulses. The samples were then diluted appropriately and characterized using nanoparticle tracking analysis. 

The dye content for each different shell was characterized by freezing each sample overnight in a −80 °C freezer, followed by lyophilization (Labconco, Kansas City, MO, USA) for a day. The resulting powder was then resuspended in 1 mL of chloroform and bath sonicated for 5 min. The resulting solution was centrifuged at 3000 RCF for 5 min. The dye concentration in the supernatant was then quantified through UV-Vis spectroscopy (Evolution 220, ThermoFisher, Waltham, MA, USA) based on a standard.

### 2.2. Cryo-Transmission Electron Microscopy Sample Preparation and Data Acquisition

Nanoparticle preparations were plunge frozen onto glow-discharged, 200 mesh copper Quantifoil grids (Quantifoil, Großlöbichau, Germany) in liquid ethane using a Vitrobot Mark IV (ThermoFisher, Hillsboro, OR, USA). Cryo-transmission electron microscopy (cryo-TEM) images were acquired in a 200 kV JEOL JEM-2200FS field emission TEM (JEOL Ltd., Tokyo, Japan) with a DE20 direct electron detector (Direct Electron LP, San Diego, CA, USA). Images were acquired with defoci of −4 µM (40 kx magnification) and −2 µM (20 kx magnification). The total dose per image ranged between 20 and 40 electrons/Å2 at magnifications of 20 k (pixel size of 2.852 Å/pixel) and 40 k (pixel size of 1.304 Å/pixel).

### 2.3. Phantom Preparation and Imaging Setup

Two polyacrylamide phantoms were constructed: one with a hollow tubular void (4.5 mm in diameter) and another with droplets embedded throughout (Figure 1). The tube phantom allowed for the nanodroplets to be suspended within water, while the polyacrylamide phantom had the nanodroplets within the polyacrylamide itself.

The transparent tissue-mimicking gel phantoms were synthesized by mixing 40% acrylamide (50 mL, Thermo Fisher Scientific, Waltham, MA, USA) with degassed water (150 mL). For the phantom with embedded droplets, nanodroplets were added to create a final concentration of 5 × 10^9^ nanodroplets per mL. Afterwards, 10% (*w*/*v*) ammonium persulfate solution (2 mL, Sigma-Aldrich, St. Louis, MO, USA) and 250 μL of TEMED (N,N,N′,N′-Tetramethylethylenediamine, Sigma-Aldrich, St. Louis, MO, USA) were added to crosslink the acrylamide. The solution was then poured into a rectangular 58 × 58 × 78 mm mold and allowed to polymerize for 30 min. To create the hollow tubular void, a plastic transfer pipette was inserted into the polymerizing solution approximately 10 mm from the top. After polymerization, the pipette was removed. Nanodroplets were diluted and placed into the tube with ultrasound gel to seal up the ends.

The phantoms were imaged from the top using an ultrasound linear array transducer L11-4v (Verasonics Inc., Kirkland, WA, USA) integrated with a 3-D printed optical fiber bundle as described previously [28]. The ultrasound transducer was driven at a center frequency of 7 MHz using a programmable research ultrasound imaging system (Vantage 128^TM^, Verasonics, Kirkland, WA, USA). The transducer was acoustically coupled to the phantom with ultrasound gel, and the optical fiber bundle was connected to a pulsed laser system (Tempest 30, New Wave Research) operating at 1064 nm. The phantom was irradiated with 5 ns laser pulses at a pulse repetition frequency of between 10–1 Hz, depending on the experiment. The output fluence was calculated based on energy measurements taken at the output of the fiber bundles using a laser power meter (Nova II, Ophir-Spiricon, North Logan, UT, USA). The nanodroplets were subjected to a pulse sequence of 10 pulses at 75 mJ/cm^2^ followed by a ramp-up in fluence, which was used to determine the threshold. The ramp-up set included one sham pulse (0 mJ/cm^2^) followed by 3 pulses at each fluence, starting at 10 mJ/cm^2^ and ending at 75 mJ/cm^2^, at intervals of approximately 10 mJ/cm^2^. This sequence was repeated a total of 10 times. The initial 10 pulses were designed to “precondition” the nanodroplets for the first cycle and were later used to simulate imaging to determine if the threshold would change over the course of imaging. The ramp-up was utilized to determine the threshold. During this process, each laser pulse was followed by 1 photoacoustic frame, followed by 100 ultrasound image frames at a 2 kHz frame rate, of which the first 5 and last 5 were saved. Each measurement was performed on six separate planes, or samples, for each condition. To determine the decay behavior, the samples were lased for 300 pulses at 75 mJ/cm^2^. Afterwards, the preconditioning pulses were removed, and the remaining points were fitted to an exponential decay function, from which the half-lives were determined. The phantoms were maintained at 37 °C in a water bath, except for the shell experiments, which were conducted at room temperature, and the core experiments, which were performed at 0 °C. This difference in temperature was due to the longer recondensation times of zonyl nanodroplets at physiological temperature as well as the non-recondensing behavior of PFP nanodroplets at room temperature. The temperature was decreased by surrounding the polyacrylamide phantom with ice for 15 min before imaging. For clarity purposes, the conditions for each experiment are described in Table 1.

**Table 1 nanomaterials-13-02238-t001:** Conditions used for each experiment.

Experiment	Phantom Type	Core	Shell	Temperature (°C)	Laser PRF (Hz)
Preconditioning (Figure 2)	Tube	PFH	Lipid	37	10
PRF Variation (Figure 3)	Polyacrylamide	PFH	Lipid	37	1, 3, 6, 10
Shell Variation (Figure 4)	Tube	PFH	Lipid, BSA, Zonyl	RT ^1^	10
Environment Variation (Figure 5)	Tube and Polyacrylamide	PFH	Lipid	37	10
Core Variation (Figure 6)	Polyacrylamide	PFP, PFH, PFHept	Lipid	0 ^2^	10

^1^ Shell variation was performed at room temperature because a significant portion of zonyl droplets did not recondense at physiological temperature. ^2^ Core variation was performed at 0 °C to allow for PFP to recondense.

### 2.4. Data Processing

Vaporization was confirmed based on ultrasound images collected after each laser pulse. Nanodroplets are generally recondensed before the fourth ultrasound frame. Differential ultrasound imaging was used to suppress the background signal and was performed by subtracting the last US frame from the first US frame after the laser pulse. Ultrasound intensity was determined by integrating over the region of interest. The vaporization threshold was determined by fitting the ultrasound intensity over the laser fluence plot with a sigmoid. Using the midpoint of the sigmoid and the slope at that point, a line was drawn, and the threshold was defined as the intersection of that line with the x axis. All data processing were performed with MATLAB. GraphPad Prism 8 (GraphPad, Boston, MA, USA) was used for plotting the figures and statistical analysis; *p* < 0.05 was considered significant.

## 3. Results and Discussion

### 3.1. Preconditioning

During the course of this study, nanodroplets were observed to exhibit an initial quiescent phase during lasing in which limited US contrast was generated. After a few pulses, the nanodroplets would begin to vaporize regularly. To demonstrate this behavior, lipid, and PFH nanodroplets in water were imaged in the tube phantom and lased repeatedly at 75 mJ/cm^2^. The initial pulse exhibits some ultrasound contrast, which increases in subsequent pulses (Figure 2A).

**Figure 2 nanomaterials-13-02238-f002:**
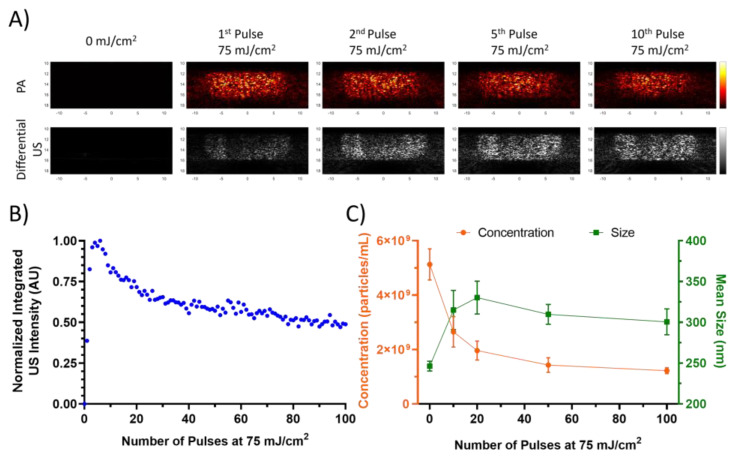
(**A**) Differential US and PA images of lipid, PFH, and nanodroplets within the tube phantom at different pulse numbers. (**B**) The normalized integrated US intensity of the same nanodroplets over the course of 100 pulses at 75 mJ/cm^2^. (**C**) Mean size and concentration taken by the NTA over the course of the 100 pulses. Error bars represent standard errors.

The increasing US contrast behavior is only observed within the first few pulses, and afterwards the nanodroplets exhibit a decrease in the integrated US intensity (Figure 2B). Size measurements taken by NTA of the nanodroplets after exposure to varying numbers of laser pulses indicate an increase in the mean size and decrease in concentration after 10 pulses, which corresponds to the approximate end of the preconditioning phase and the start of the decay phase. This suggests that the initial lasing results in the destruction of smaller nanodroplets that may coalesce to form larger nanodroplets. Similar behavior has been observed optically in ADV studies on larger droplets, where the resulting microbubble would increase in diameter after repeated cycles [20]. Such an occurrence would explain the sudden increase in US contrast within the initial pulses as the nanodroplets coalesce from laser perturbation. The subsequent decay is likely due to the more gradual destruction of the nanodroplets without as much coalescence, as evidenced by the decrease in concentration without a substantial change in the mean size. 

This process could explain the absence of “preconditioning” behavior observed in a study by Namen et al., which utilized ADV for repeated vaporization of PFH nanodroplets [29]. In their study, a 10 cycle ultrasound pulse at 1.1 MHz was used to trigger the transient phase change. Nanodroplet coalescence most likely occurred during the initial cycles. Since the resulting microbubbles were only imaged after the activation pulse, no “preconditioning” behavior was observed. Moreover, it is important to note that in ADV, energy is deposited over a longer period of time. In the study by Namen et al., the pulse duration was nine microseconds, while this investigation employed the use of a five nanosecond pulsed laser. We theorize that the longer duration of the high intensity focused ultrasound (HIFU) exposure, along with its mechanical impact, facilitates the coalescence of the nanodroplets during the initial activation pulse. In the case of ODV, heat is deposited in a shorter period, likely requiring more pulses to achieve “preconditioning”.

Although this behavior may impose some limitations on imaging these particles in the body, there are straightforward methods to overcome them. One approach is to “precondition” the nanodroplets with several pulses prior to injection, ensuring ideal behavior in situ. However, this may change the size distribution and impact on trafficking of the nanodroplets. Alternatively, additional pulses can be incorporated into the imaging sequence that can allow for “preconditioning” in vivo without altering nanodroplet size prior to injection. On a side note, this unique behavior could be leveraged to enhance nanodroplet identification in situ.

### 3.2. PRF Variation

Another possible contributor to the “preconditioning” behavior could be the buildup of heat over lasing. This could be explored by altering the laser PRF. By increasing the time between each laser pulse, there would be more time for heat to diffuse.

Previous work has shown that acquisition settings such as imaging pulse polarity can influence nanodroplet behavior, and HIFU along with laser irradiation can reduce the vaporization threshold [17,30]. To our knowledge, no studies have examined the impact of the laser itself. To test the impact of the laser PRF on the nanodroplet performance, PFH and lipid nanodroplets were embedded in a polyacrylamide phantom. This prevents the nanodroplets from diffusing in and out of the lasing plane, ensuring consistent exposure to the same population of nanodroplets. The impact of the polyacrylamide phantom is further discussed in a later section. In order to determine the vaporization threshold, a ramp cycle composed of three pulses at each fluence ranging from 0 to 75 mJ/cm^2^ at intervals of approximately 10 mJ/cm^2^ was used. These cycles were interleaved between sequences of 10 pulses at 75 mJ/cm^2^. This was done to initially precondition the nanodroplets and further simulate imaging of the nanodroplets to determine how the threshold changes over repeated lasing. Nanodroplets were lased at a variety of PRFs (1 Hz, 3 Hz, 6 Hz, and 10 Hz) and found to have a small impact on the vaporization threshold (Figure 3).

**Figure 3 nanomaterials-13-02238-f003:**
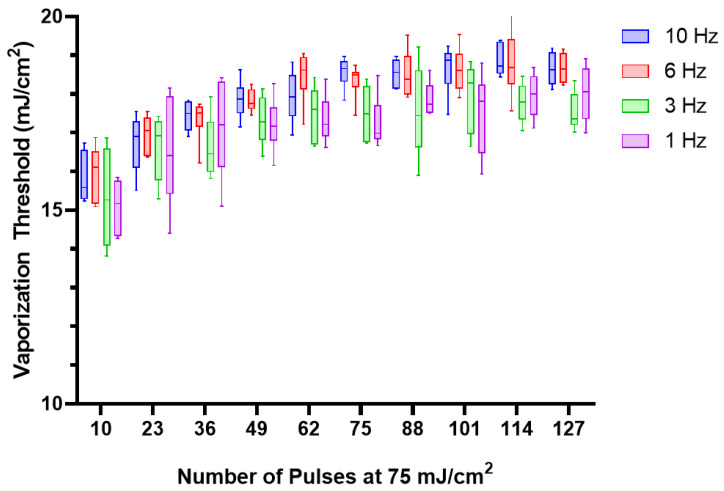
The vaporization threshold of lipid, PFH, nanodroplets (n = 6) embedded in a polyacrylamide phantom at different laser pulse repetition frequencies (PRF; 10 Hz, 6 Hz, 3 Hz, and 1 Hz). The different PRFs did exhibit a statistical difference (repeated measures ANOVA; F(3, 20) = 4.146, *p* = 0.0194), with differences detected between 10 Hz and 3 Hz/1 Hz (*p* = 0.0014 and *p* = 0.0029) and 6 Hz and 3 Hz/1 Hz (*p* = 0.0005 and *p* = 0.0012). However, the mean differences were small (~0.8 mJ/cm^2^).

This result suggests that, based on the PRFs used in this study, heat buildup is unlikely, as that would result in a reduced vaporization threshold for the nanodroplets lazed at higher PRFs. Oddly enough, the nanodroplets lazed at 10 Hz and 6 Hz exhibited a slightly elevated vaporization threshold (~0.8 mJ/cm^2^) in comparison to the 3 Hz and 1 Hz samples. The difference in vaporization threshold is relatively small and likely an artefact of the laser itself due to pulse-to-pulse variations at different PRFs, which can vary by 2.5%. These slight differences could contribute to the difference between the PRFs.

Studies examining PRF for ADV performed by Wu et al. and Fabiilli et al. showed that higher PRFs result in lower pressure thresholds for lipid-coated perfluorobutane nanodroplets [20,21]. However, these studies were performed in a flow environment, and increasing the PRF would increase the number of pulses to which a subpopulation of nanodroplets were exposed. Thus, it is not analogous to compare the impact of laser PRFs on ODV thresholds to the impact of ultrasound PRFs on ADV.

### 3.3. Shell Variation

Previous studies have used a variety of materials to stabilize droplets, but no work has been performed directly comparing shell compositions and their effect on ODV. Typically, droplets are synthesized with a lipid shell using DSPC-PEG, DSPE, or similar lipids. Others have used proteins such as bovine serum albumin (BSA), fluorosurfactants such as zonyl, krytox, or other polymers to stabilize the droplet [3,9,23,30,31,32,33,34,35]. In lipid nanodroplets, shell composition has been shown to impact the echogenicity and size of the droplets [26,36]. This would suggest that changing the shell composition could also impact the vaporization threshold and recondensation of the nanodroplets.

To examine this hypothesis, nanodroplets formulated with lipid, BSA, and zonyl, a fluorosurfactant, were characterized at the same concentration (5 × 10^9^ nanodroplets/mL) in the tube phantom. The different formulations were of similar sizes (z-average size BSA: 314 nm, lipid: 293.6 nm, and zonyl: 285.8 nm). These experiments were performed at room temperature in water, as zonyl nanodroplets did not fully recondense at physiological temperature before the subsequent laser pulse.

The different formulations had a substantial impact on the vaporization threshold (Figure 4A), with zonyl nanodroplets exhibiting the lowest vaporization threshold, followed by lipid and then BSA. However, each formulation contained different amounts of dye (Figure 4B). Surprisingly, the dye content did not correlate with the vaporization threshold, with BSA having the most, followed by lipid, and then zonyl. This suggests that the shell has an impact on the vaporization threshold independently of the dye content. 

**Figure 4 nanomaterials-13-02238-f004:**
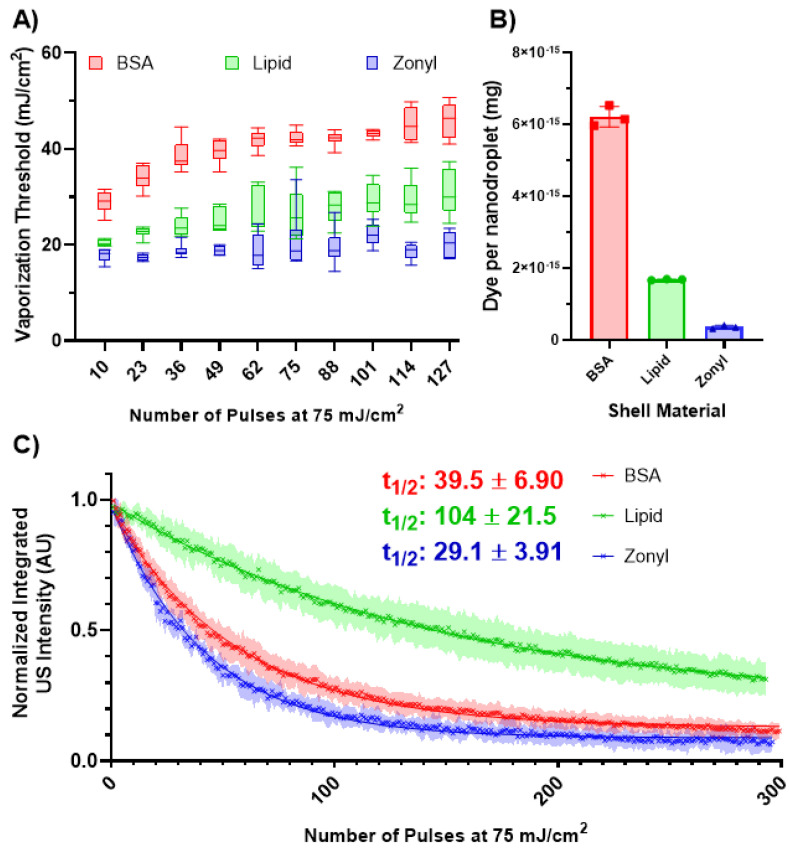
(**A**) The vaporization threshold of the different formulations of the nanodroplets (n = 6) after exposure to varying numbers of laser pulses (75 mJ/cm^2^). Significant differences in the vaporization threshold were observed among the different shells (F(2, 15) = 172.3, *p* < 0.0001). Tukey’s comparison test indicated significant differences between all shells (*p* < 0.0001). (**B**) Dye content per formulation (n = 3). Error bars represent the standard deviation. (**C**) The decay behavior of the normalized integrated US intensity for each formulation (n = 6) after repeated lasing at 75 mJ/cm^2^. The half-life for each formulation was determined by fitting each curve with an exponential decay.

The difference in vaporization thresholds could be due to the difference in the interfacial tension at the water/PFH interface. Lipid surfactants have been shown to have minimal interfacial tension between water and perfluorocarbons, with measurements at 1.14 dynes/cm and the interfacial tension between water and PFP stabilized by zonyl was found to be 5.8 dynes/cm [37,38]. The interfacial tension of BSA has been reported to be ~30 dynes/cm, which is much higher than that of lipids and zonyl [39]. This would result in BSA exhibiting a Laplace pressure of 400 kPa, compared to 15.2 kPa and 77.3 kPa for lipids and zonyl, respectively. Approximating the boiling temperature utilizing the Antoine equation [40,41] results in a boiling temperature of 132.85 °C, 61.9 °C, and 77.65 °C for BSA, lipid, and zonyl, respectively. While these differences do appear to somewhat agree with the vaporization thresholds, the dye loading values suggest that the threshold for BSA should be far greater, and like for ADV, Laplace pressure alone is unlikely to predict the vaporization threshold for ODV [42].

Another factor that could influence the vaporization threshold could be the intermolecular forces between the surfactants. A study performed by Mountford et al. demonstrated that increasing acyl chain length increased the activation energy required for ADV [43], and Huang et al. showed that crosslinking polymers on the surface of the perfluorocarbon nanodroplet increased the ADV threshold compared to uncrosslinked polymers [44]. Since BSA microbubbles are stabilized by cross-linked cysteine residues found on the proteins formed due to cavitation during the sonication process [45], this suggests that higher vaporization thresholds for BSA nanodroplets could be due to crosslinking of the BSA proteins. Moreover, fluoroalkanes have weaker intermolecular forces in comparison to their hydrocarbon cousins [46]. This could explain the difference in the vaporization thresholds between zonyl and lipid nanodroplets, as the hydrophobic tails of the lipids have a higher activation energy compared to the fluorinated segment of the fluorosurfactants.

In order to better isolate the impact of the shell, future studies should conjugate the photoabsorber onto the surface of different shells at known ratios such that concentrations will be uniform between samples. 

Following repeated lasing at 75 mJ/cm^2^, all the formulations exhibited a decay in US contrast (Figure 4C). This decay is unlikely to be driven by photobleaching of the dye, as epolight 3072 appears to be stable past 400 pulses at 90 mJ/cm^2^, showing little change in the UV-Vis spectrum (S1). This suggests that the decrease in US contrast is primarily driven by nanodroplet destruction or ejection of the dye.

Lipid nanodroplets exhibited the longest half-life (100 pulses) in comparison to BSA and zonyl, which had half-lives of 39.5 and 29.1 pulses, respectively. The longevity of the nanodroplets may be related to the flexibility of their shells. Lipid microbubbles exhibit more flexibility in their shell compared to the more rigid BSA, which may explain the ability of these nanodroplets to undergo repeated vaporization events [47]. However, there are few studies that examine the shell behavior of fluorosurfactants, limiting any further comparisons.

After vaporization, the distribution of the surfactants could also influence the nanodroplet’s performance. A comparison between nanodroplets formulated from fluorescently tagged lipids and BSA showed that the lipids were homogeneously distributed after vaporization, while the BSA bubble exhibited a heterogeneous coating. Furthermore, lipid nanodroplets exhibited slower expansion post-vaporization in comparison to BSA and fluorosurfactant-stabilized nanodroplets, suggesting that lipids were better retained and limited gas transfer [34]. These factors could limit the fragmentation of the shell, loss of dye, and diffusion of PFH out of the nanodroplet, resulting in more sustainable vaporizations. 

### 3.4. Environmental Variation

To determine the impact of the local environment on the nanodroplet vaporization, lipid nanodroplets were embedded in polyacrylamide and compared to those suspended in water within the tube phantom at the same concentration (5 × 10^9^ nanodroplets/mL). The vaporization thresholds were initially similar (Figure 5A), but they quickly diverged. The initial similarity suggests that embedding the nanodroplets within the phantom had minimal impact on the nanodroplet vaporization. 

Further lasing leads to an increase in the average and standard deviation of the threshold of the nanodroplets in water, whereas in the phantom, the increase is more gradual with smaller standard deviations. This observation suggests that nanodroplets with lower vaporization thresholds were destroyed in water, while the phantom environment likely mitigated nanodroplet destruction. This is supported by Figure 5B, where the nanodroplets are lased repeatedly at 75 mJ/cm^2^ and demonstrate a limited decrease in US contrast in comparison to nanodroplets in water.

**Figure 5 nanomaterials-13-02238-f005:**
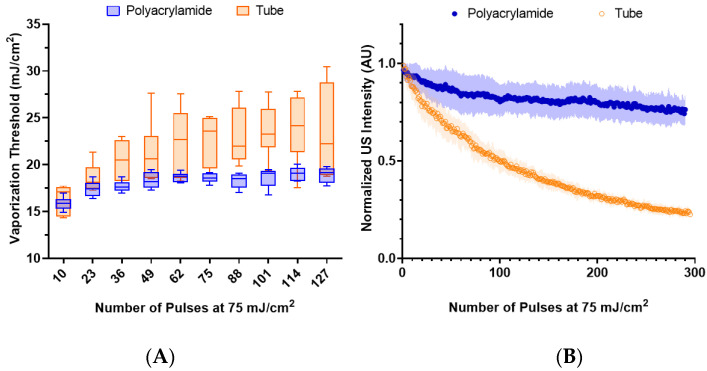
(**A**) The vaporization threshold of lipid, PFH, nanodroplets (n = 6) embedded in a polyacrylamide phantom compared to those within the tube phantom over repeated lasing at 75 mJ/cm^2^. The environments exhibited a statistically significant difference (paired *t*-test, *p* < 0.0001). (**B**) The decay behavior of the normalized integrated US intensity for each environment (n = 6) after repeated lasing (75 mJ/cm^2^).

Microbubble behavior is known to be impacted by the surrounding environment, where fragmentation of microbubbles is reduced in more viscous fluids [48]. Thus, it is likely that the viscoelastic properties of the polyacrylamide phantom help stabilize the vaporization of the nanodroplets, reducing their destruction.

Furthermore, it was observed that repeated vaporization of PFH nanodroplets in the phantom resulted in the formation of residual microbubbles that did not recondense within the duration of the imaging sequence. This behavior has also been observed in repeated ADV of lipid PFH nanodroplets [29]. The nanodroplets likely coalesced together during the repeated vaporization, eventually forming stable microbubbles. A similar behavior has been described with ADV of PFP nanodroplets, where repeated vaporization is needed to result in the formation of stable microbubbles [49]. Meanwhile, nanodroplets in water did not exhibit this behavior.

These observations have implications for characterizing nanodroplet behavior. Nanodroplet behavior is commonly studied within phantoms such as polyacrylamide or agarose. While this does allow for observation of individual nanodroplet dynamics without movement of the particles, it’s clear that the phantom could help stabilize the nanodroplets and resulting microbubbles, and the impact of the phantom should be considered.

Surface wave elastography was used to measure the stiffness of the polyacrylamide phantom, which was found to have a Young’s modulus of 100 ± 3.11 kPa (Appendix A). This is notably higher than most tissues within the human body [50], suggesting that nanodroplet vaporization is unlikely to be suppressed due to the stiffness of the surrounding tissue. This observation holds implications for applications where the nanodroplets are embedded in tissue, bound to a cancer cell [51], or taken up by macrophages [52,53,54]. 

However, the mean peak ultrasound intensity showed a 46% reduction in the polyacrylamide in comparison to the tube phantom. This could be due to two main factors: destruction of nanodroplets during polymerization and suppression of microbubble expansion. The polyacrylamide polymerization process is exothermic, which could result in the destruction of the nanodroplets through spontaneous vaporization from heating. Microbubbles can sometimes be visualized before the lasing of the phantom (Figure 1). Furthermore, the polyacrylamide phantom itself can restrict the expansion of the microbubble.

Repeated lasing of the phantom was not observed to impact the polyacrylamide phantom. Photodegradation of hydrogels primarily occurs through the process of plasma generation at the site of lasing, which results in the formation of bubbles that cavitate [55]. While the nanodroplets embedded in the phantom undergo a similar process, these particles occupy a small volume fraction of the phantom (~0.004%), and no damage was observed visually or in the ultrasound images. 

### 3.5. Core Variation

Previous work has shown that ADV thresholds heavily depend on the boiling temperature of the core [5,56]. However, it is unclear how transitory ODV thresholds of sub-micron-sized nanodroplets depend on boiling temperature, as previous work utilized micron sized droplets, validated vaporization optically, or only characterized one core material [4,23,30].

To this end, lipid nanodroplets were synthesized with PFP, PFH, and PFHept with z-averaged sizes of 309.0 nm, 293.6 nm, and 265.1 nm, respectively. These nanodroplets were then embedded within polyacrylamide phantoms to ensure more precise measurements of the vaporization threshold. The phantoms were chilled to 0 °C to allow for repeated vaporization of PFP. The vaporization thresholds show variations based on the core material (Figure 6A). The average vaporization thresholds of the nanodroplets were plotted against the critical temperature of their respective cores and fitted with a linear regression, assuming vaporization occurred via homogeneous nucleation, with an x-intercept at 0 °C (Figure 6B). This suggests that ODV thresholds are correlated to the critical temperature, and deviations could be due to minor variations in concentration, size, or pulse-to-pulse variation in laser fluence. However, application of classical homogeneous nucleation theory may not present the most accurate depiction of ODV as it does not account for interfacial tension, which our results suggest can impact the threshold (Figure 4A) and the photoacoustic pressure generated from the dye itself. Overall, the observed behavior follows trends observed in ADV [20,22] and previous observations performed by Dove et al. [4].

**Figure 6 nanomaterials-13-02238-f006:**
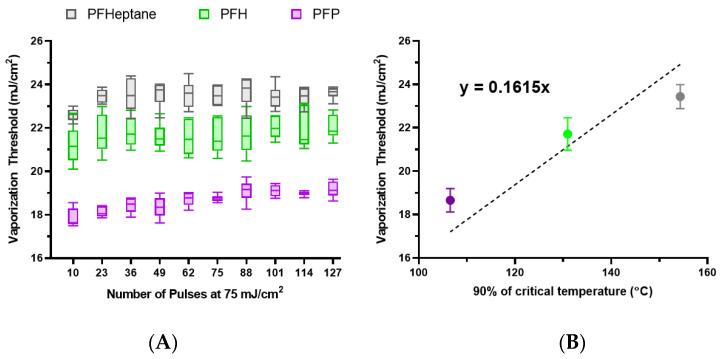
(**A**) The vaporization thresholds of lipid nanodroplets with different cores embedded in a polyacrylamide phantom that has been cooled to 0 °C. (**B**) The averaged vaporization thresholds for each core are plotted against 90% of the critical temperature for each core and fitted with a linear fit with an x-intercept set to 0 °C. Error bars represent the standard deviation.

The observed difference in transient vaporization thresholds is substantially lower than that previously reported in optical vaporization measurements [4,14,30]. Previous work noted substantial differences between the thresholds of PFP and PFH. This implies that the vaporization threshold could depend on the formulation of the particle. Dove et al. formulated perfluorobutane (boiling point: −1.7 °C) droplets with gold nanoparticles conjugated onto the lipid shell and found the vaporization threshold to be 26.1 ± 21.7 mJ/cm^2^, which is similar to the vaporization threshold for the lipid PFH (boiling point: 56 °C) nanodroplets in our study. Though not directly comparable due to different absorbers and cores, this result suggests that embedding the optical absorber within the droplet shell allows for higher efficiencies of energy transfer that result in lower vaporization thresholds. Surface conjugation can result in lower amounts of absorber per droplet due to steric hinderance and limited binding sites, while encapsulation within the lipid membrane can lead to the incorporation of greater amounts of dye and higher heat transfer efficiencies compared to surface-bound nanoparticles. This is further supported by work done by Wei et al., where gold nanospheres coated with hydrophobic and hydrophilic ligands were used to create the shell, resulting in vaporization thresholds as low as 3.5 mJ/cm^2^ for PFH nanodroplets [57].

In this study, several factors that impact the vaporization threshold have been identified; however, to fully understand the underlying mechanisms of ODV, other factors that impact Laplace pressure should be examined, such as nanodroplet size. Future studies can utilize microfluidic devices to either sort or synthesize monodisperse nanodroplets to systematically examine the impact of size on nanodroplet vaporization, which would help elucidate the interplay between Laplace pressure and ODV. 

## 4. Conclusions

Overall, this study has investigated the dynamics of repeated vaporization of high boiling point PFCs and identified a short phase in which the nanodroplets exhibit increased US contrast, which we term “preconditioning”. This process is likely due to a change in the size of the nanodroplet and not to the buildup of residual heat. We further examined the relationship between ODV thresholds and different shell materials, environments, and perfluorocarbon cores. Nanodroplet shells are shown to have an impact on both the vaporization threshold and imaging half-life. ODV is shown to not be impacted by environmental stiffness, and the vaporization threshold of high-boiling-point perfluorocarbons correlates with the critical temperature. Overall, this work provides a foundation for future engineering of ODV nanodroplets and a starting point for exploring other properties of the nanodroplet, such as the “preconditioning” behavior and the development of a theoretical model for ODV. These advances could lead to improved use of these contrast agents for photoacoustic and ultrasound imaging and image-guided therapy.

## Data Availability

Data can be made available upon request.

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
