# Peer review of "Factors Influencing the Repeated Transient Optical Droplet Vaporization Threshold and Lifetimes of Phase Change, Perfluorocarbon Nanodroplets"

_nanomaterials, 2023, doi:10.3390/nano13152238_

Round 1
Reviewer 1 Report
The manuscript "Factors Influencing the Repeated Transient Optical Droplet Vaporization Threshold and Lifetimes of Phase-change, Perfluorocarbon Nanodroplets" presents a study on nanodroplets made of perfluorinated oils stabilized by various amphiphiles in either water or a polymer. The droplets were dynamically evaporated using laser pulses (absorption was ensured by dyes). Typical characteristics of the process were monitored. The process seems to be reproducible. However the presentation has some flaws that should be corrected.
(1) The meaning of this process in medical applications did not at all become clear to me. Why should it be important here?
(2) The droplet size was kept constant (mainly due to the stability of the emulsions. If the authors had used microemulsions the droplet size could have been varied systematically by the amount of amphiphile - at least by the zonyl. This important aspect is missing.
(3) For medical applications I would rather expect gels to be important. Then the analysis of the tissue material after light exposure becomes important. What could be learned here?
Reviewer 2 Report
The study titled "Factors Influencing the Repeated Transient Optical Droplet Vaporization Threshold and Lifetimes of Phase-change, Perfluorocarbon Nanodroplets" presents intriguing research that addresses the necessity of comprehending vaporization dynamics in smaller nanodroplets through appropriate techniques. This manuscript exhibits commendable qualities and aligns well with the scope of Nanomaterials, making it suitable for publication. While the manuscript only requires minor revisions before publication, there are a few areas that would benefit from improvement.
Firstly, the introduction section would benefit from additional background information elucidating the applications of nanodroplets. Expanding upon the practical uses and significance of nanodroplets in various fields will provide readers with a better understanding of the context and importance of the study.
Furthermore, the supplementary material section requires substantial organization. It is recommended to assign numerical labels to all sections and provide comprehensive captions for all elements, including tables and figures. This will enhance clarity and facilitate ease of navigation for readers. Additionally, it would be advantageous to allocate a separate section specifically for the methods, ensuring a more streamlined and structured presentation.
Round 2
Reviewer 1 Report
The authors replied to all my points. So I promote the publication.